# High-Fat-Diet-Evoked Disruption of the Rat Dorsomedial Hypothalamic Clock Can Be Prevented by Restricted Nighttime Feeding

**DOI:** 10.3390/nu14235034

**Published:** 2022-11-26

**Authors:** Anna Magdalena Sanetra, Katarzyna Palus-Chramiec, Lukasz Chrobok, Jagoda Stanislawa Jeczmien-Lazur, Emilia Gawron, Jasmin Daniela Klich, Kamil Pradel, Marian Henryk Lewandowski

**Affiliations:** 1Department of Neurophysiology and Chronobiology, Institute of Zoology and Biomedical Research, Jagiellonian University in Krakow, Gronostajowa Street 9, 30-387 Krakow, Poland; 2School of Physiology, Pharmacology, and Neuroscience, University of Bristol, University Walk, Biomedical Sciences Building, Bristol BS8 1TD, UK; 3Max-Delbrück-Center for Molecular Medicine in the Helmholtz Association (MDC), Robert-Rössle-Str. 10, 13125 Berlin, Germany

**Keywords:** chronobiology, metabolism, time-restricted feeding, obesity, food-entrainable oscillator

## Abstract

Obesity is a growing health problem for modern society; therefore, it has become extremely important to study not only its negative implications but also its developmental mechanism. Its links to disrupted circadian rhythmicity are indisputable but are still not well studied on the cellular level. Circadian food intake and metabolism are controlled by a set of brain structures referred to as the food-entrainable oscillator, among which the dorsomedial hypothalamus (DMH) seems to be especially heavily affected by diet-induced obesity. In this study, we evaluated the effects of a short-term high-fat diet (HFD) on the physiology of the male rat DMH, with special attention to its day/night changes. Using immunofluorescence and electrophysiology we found that both cFos immunoreactivity and electrical activity rhythms become disrupted after as few as 4 weeks of HFD consumption, so before the onset of excessive weight gain. This indicates that the DMH impairment is a possible factor in obesity development. The DMH cellular activity under an HFD became increased during the non-active daytime, which coincides with a disrupted rhythm in food intake. In order to explore the relationship between them, a separate group of rats underwent time-restricted feeding with access to food only during the nighttime. Such an approach completely abolished the disruptive effects of the HFD on the DMH clock, confirming its dependence on the feeding schedule of the animal. The presented data highlight the importance of a temporally regulated feeding pattern on the physiology of the hypothalamic center for food intake and metabolism regulation, and propose time-restricted feeding as a possible prevention of the circadian dysregulation observed under an HFD.

## 1. Introduction

Overweight and obesity have reached pandemic proportions, with almost two billion overweight and more than half a billion obese adults worldwide as of 2016 [1]. A high body mass index (BMI) is a risk factor for many life-threatening disorders, such as cardiovascular disease [2] and various types of cancer [3], which are the two leading causes of death globally. Although the pathophysiology of obesity is complex and multifactorial, it is considered to be mostly the result of an imbalance between the number of calories ingested and spent, followed by the accumulation of the excess in the form of fat tissue. Modern dietary habits promote a disproportionate intake of highly palatable foods, rich in fat and/or sugar, increasing the hedonic motivational drive and disturbing the homeostatic balance.

A relatively new approach to metabolism research involves its circadian regulation. An increased BMI is often associated with shift work [4,5,6,7,8], which disturbs the body clock by its irregular activity and feeding patterns [9,10], raising the question of how much the disruptions in feeding patterns influence weight gain prevalence. Time-restricted feeding with daily periods of food deprivation have been shown to be beneficial for losing weight and improving sleep quality in humans [11], as well as to prevent excessive weight gain in mice fed a high-fat diet (HFD) [12,13,14] and enhance longevity in rodents [15].

An HFD is the most commonly used animal model of diet-induced obesity (DIO). In addition to body weight gain, it induces leptin resistance [16] and type 2 diabetes, and impairs glucose tolerance [17]. Interestingly, an HFD was also shown to lengthen the circadian period and disrupt the day/night feeding pattern even before obesity development [18,19], further confirming circadian dysregulation as an important factor in its pathophysiology.

The circadian rhythmicity of different functions is controlled by local oscillators, present in the brain as well as different tissues on the periphery. These clocks are synchronized by the light-entrained master circadian clock in the suprachiasmatic nucleus (SCN) of the anterior hypothalamus, which can, however, be uncoupled from its metabolic partners by an unhealthy feeding pattern [20]. The brain structures sensitive to metabolic cues and orchestrating behavioral responsiveness to different feeding schedules collectively form the food-entrainable oscillator (FEO), of which a very important part is the dorsomedial hypothalamus (DMH) [21]. The DMH is sensitive to many hunger and satiety signals, including ghrelin [22,23,24], orexins [25,26], cholecystokinin [24,27], leptin [28,29,30,31,32] and glucagon-like peptides [33,34], involved in both short- and long-term regulation of feeding behaviour. Moreover, the DMH exhibits an intrinsic circadian rhythm in clock gene expression [35], which can be enhanced and even entirely shifted by restricted feeding [36,37,38,39,40].

DIO has been shown to globally affect the DMH, changing the expression of over 80 genes in this structure [41], which is more than in any other investigated brain structure (importantly including other hypothalamic metabolic centers such as the lateral hypothalamus, the paraventricular nucleus or the arcuate nucleus). However, it is not clear when these changes begin, and whether they are a result of obesity or contribute to its development by affecting feeding behaviour and metabolism. Therefore, in this study, we explored how a short-term (2–4 weeks) HFD (preceding obesity, [19]) influences day/night rhythms in this hypothalamic structure and how this relates to the disrupted pattern of food intake observed under an HFD. Using ex vivo electrophysiology and immunofluorescence we show that even such a short period of an HFD consumption is enough to impair DMH cellular activity rhythms, in the form of an increasing daytime cFos immunoreactivity and neuronal firing rate, as well as by delaying the peak of its electrical activity. Most importantly, these disruptions do not occur under restricted nighttime feeding (NF), indicating their possible prevention by a healthy feeding pattern.

## 2. Materials and Methods

### 2.1. Ethical Statement

All experiments were performed in accordance with the Polish Animal Welfare Act of 23 May 2012 (82/2012) and the European Communities Council Directive (86/609/EEC) and were approved by the Local Ethics Committee in Krakow (No. 18/2018, 349/2022). Every possible effort was made to minimize the number of animals used and their suffering.

### 2.2. Animal Maintenance

The study was performed on male Sprague-Dawley (SD) rats, bred and kept at the Jagiellonian University in Krakow (Poland) in standard lighting conditions (LD 12/12), constant temperature (23 ± 2 °C) and humidity (~65%), with water and food supplied ad libitum (with the exception of the night-feeding protocols). Females were excluded due to probable interactions between the estrous phase and food intake/metabolism, especially during puberty. At 4 weeks of age, the animals were weaned and assigned to either control or experimental group. The first one was fed control diet (CD; ~3514 kcal/kg, fat content 4%, energy from: 10% fat, 24% protein, 66% carbohydrates, cat. no. C1090-10; Altromin International, Lage, Germany), whereas the experimental group received HFD (~5389 kcal/kg, fat content 42%, energy from: 70% fat, 16% protein, 14% carbohydrates, cat. no. C1090-70; Altromin International). The animals were then fed a respective diet for 2–7 weeks, depending on the experimental protocol, as detailed in the following sections.

### 2.3. Food Intake and Body Weight Gain Assessment

Food intake and body weight gain assessment was performed on 20 rats (10 of which fed CD and another 10 HFD). In order to measure the amount of food eaten, the rats were housed individually. Starting 3 days after weaning, body weight and the amount of food eaten during 24 h were measured every week for 7 successive weeks, always at the beginning of the light phase (ZT—Zeitgeber time 0). Food ingested was analysed both as the chow mass disappearing from the feeder, as well as after calculating this value into kcal/kg of body weight.

### 2.4. Immunofluorescence

For the immunofluorescence study, 50 animals were used (24 fed CD and 26 fed HFD). After 4 weeks on either diet, the rats were perfused at one of 4 time points (ZT0, 6, 12 or 18). Each of the new groups formed contained 6 animals, with the exception of HFD—ZT6 and HFD—ZT18, which contained 7 rats.

All animals were anesthetized by isoflurane inhalation (1 mL in the incubation chamber, Baxter, Deerfield, IL, USA; *v*/*v* air mixture) followed by sodium pentobarbital injection (100 mg/kg body weight, i.p.; Biowet, Pulawy, Poland). Deep anesthesia was confirmed with a tail pinch, and when no response was observed, the rats were perfused transcardially with phosphate-buffered saline (PBS) followed by 4% paraformaldehyde (PFA) in PBS. Next, the brains were removed and kept in the same PFA solution overnight. A vibroslicer (Leica VT1000S, Heidelberg, Germany) was used to cut 35 μm thick slices containing the DMH, which were then washed out of PFA by two 10 min long incubations with PBS. Non-specific site blocking and membrane permeabilization were performed in one step, with a 30 min long incubation in a PBS solution containing 0.6% Triton-X100 (Sigma-Aldrich, Saint Louis, MO, USA) and 10% normal donkey serum (NDS, Jackson ImmunoResearch, West Grove, PA, USA) at room temperature. Next, the slices were transferred to a PBS solution containing rabbit anti-cFos antibodies (1:2000, Abcam, Cambridge, UK), 2% NDS and 0.3% Triton-X100 and kept in it for 24 h at 4 °C. After this step, the slices were rinsed in PBS (2 × 10 min) and incubated with secondary antibodies (AlexaFluor488-conjugated anti-rabbit antisera; 1:400, Jackson ImmunoResearch, West Grove, PA, USA) in PBS overnight at 4 °C. Finally, the slices were rinsed again (2 × 10 min), mounted onto glass slides and coverslipped with FluoroshieldTM with DAPI (Sigma-Aldrich, Saint Louis, MO, USA).

Slices were scanned using epifluorescence microscope (Axio Imager M2, Zeiss, Jena, Germany) at 20× magnification, cFos-positive cells were counted manually and DMH area measured with the aid of ZEN 2.5 (blue edition) software (Zeiss, Jena, Germany). Only cells immunoreactive for cFos and DAPI-positive were counted. Results present the density of cFos-positive cells within the area occupied by the DMH.

### 2.5. Electrophysiology

#### 2.5.1. Short-Term Recordings

##### Tissue Preparation

Multi-electrode array (MEA) ex vivo recordings were performed in the middle of both light and dark phases. A total of 16 animals were used (CD daytime: 5, HFD daytime: 5, CD nighttime: 3, HFD nighttime: 3), fed with a respective diet for 2–4 weeks. Animals were sacrificed between 1–3 h after the onset of each lighting phase (ZT1–3 and ZT13–15), and the recordings were performed around ZT6/18 (±1 h).

Rats were anesthetized with isoflurane (2 mL/kg body weight), then decapitated. The brain was removed while immersed in ice-cold, cutting artificial cerebro-spinal fluid (cACSF), containing (in mM): 25 NaHCO_3_, 3 KCl, 1.2 Na_2_HPO_4_, 2 CaCl_2_, 10 MgCl_2_, 10 glucose, 125 sucrose with addition of a pH indicator, Phenol Red 0.01 mg/L, osmolality ~290 mOsmol/kg, continuously carbogenated (95% O_2_, 5% CO_2_). Then, 250 µm thick coronal slices containing the DMH were cut using a vibroslicer (Leica VT1000S, Heidelberg, Germany) and incubated in the recording artificial cerebro-spinal fluid (rACSF), containing: (in mM): 125 NaCl, 25 NaHCO_3_, 3 KCl, 1.2 Na_2_HPO_4_, 2 CaCl_2_, 2 MgCl_2_, 5 glucose and 0.01 mg/l of Phenol Red (initial temperature: 32 °C, cooled to room temperature) for a minimum of 0.5 h before the recording.

##### Multi-Electrode Array Recordings

The MEA recordings were performed using MEA2100-System (Multichannel Systems GmbH, Reutlingen, Germany; [42]). Slices containing the DMH were placed onto an 8 × 8 recording array of a perforated MEA (60pMEA100/30iR-Ti, Multichannel Systems), constantly perfused with fresh rACSF, heated up to 32 °C. The slices were positioned to ensure the presence of as many recording spots within the DMH as possible, and then gently sucked into the perforations neighboring the spots by creating imbalance between inward and outward flow through the bottom tubing circuit of the recording chamber. After a proper amount of suction was established, the slices were allowed to settle for half an hour before the start of the recording and were then recorded for another half an hour.

Signal was acquired with Multi Channel Experimenter software (Multichannel Systems), with sampling frequency of 20 kHz. After the experiment, the raw signal was processed as described previously [19,43]. In brief, the signal was exported to HDF5 and CED-64 files with Multi Channel DataManager (Multichannel Systems GmbH). The HDF5 file was mapped and converted into DAT format with a custom-made MatLab script (R2018a version, MathWorks, Natick, MA, USA), followed by an automatic spike sorting with the KiloSort [44] in MatLab environment. Parallelly, the CED-64 files were remapped and filtered with Butterworth band pass filter (fourth order) from 0.3 to 7.5 kHz by a custom-made Spike2 script. Spike-sorting results were then transferred into the prepared CED-64 files (Spike2 8.11; Cambridge Electronic Design Ltd., Cambridge, UK) with a custom-written MatLab script. At the end, each spot was verified manually for errors in the automatized process, and corrected with the aid of autocorrelation, cross-correlation, principal component analysis (PCA) and spike shape inspection.

#### 2.5.2. Long-Term Recordings

For long-term MEA experiments, animals (3 rats per diet) were sacrificed at ZT0, and the recordings started exactly 2 h later. The procedures for both tissue preparation, MEA setup and signal analysis were analogical to those reported in the previous section (2.5.1 Short-Term Recordings) with two differences. First, as the recordings lasted ~30 h, they were not recorded in a continuous way, but one-minute-long sample was recorded every ten minutes. Second, to enhance long-term survival of the tissue, 1 mg/mL of penicillin–streptomycin (Sigma-Aldrich) was added to the rACSF and the recordings were performed at 25 °C.

### 2.6. Night-Feeding Protocol

After the administration of a respective diet (CD or HFD, at weaning) the animals were fed ad libitum for the first two weeks, after which they were only given access to food during nighttime, for another 2 weeks. Following this, two separate experiments were performed: the long-term MEA recordings, analogically to the procedure described earlier (2.5.2 Long-Term Recordings; 4 rats fed CD and 3 fed HFD), and a combination of short-term MEA recordings with an immunofluorescence study, as follows.

#### Short-Term MEA Recordings Combined with Immunofluorescence Staining

For these experiments, we used a total of 16 rats (groups included: CD daytime, CD nighttime, HFD daytime, HFD nighttime, 4 animals per group). Tissue was collected as reported in the short-term MEA recordings (2.5.1) section and the MEA experiments performed accordingly to the study on ad libitum-fed animals. From each animal one 250 µm thick slice containing DMH was used for the MEA, and the rest was immediately fixed in a 4% PFA solution overnight. The latter slices were then immersed in a sucrose solution (30% in PBS) for ~24 h, cut into thinner, 35 µm slices using a cold-plate microtome, and further processed as described in the Immunofluorescence (2.4) section. For this experiment the animals were sacrificed at ZT3–4/ZT15–16, to ensure that both slice fixation in the PFA solution, as well as the MEA recordings, happened around ZT6/ZT18 (±1 h).

### 2.7. Statistical Analysis

Statistical analysis was performed in R (Version 4.0.4; [45]) and RStudio (Version 1.4.1106, PBC; [46]). Outliers were detected with the help of the box-and-whisker method from the rstatix package [47] and those identified as extreme outliers (above Q3 + 3×IQR or below Q1—3×IQR) were removed from further analysis. For independent data, a general linear model was fitted, whereas in the case of multiple observations from the same animals, a random intercept was included in a linear mixed effects model (and a random slope for repeated measures designs). Mixed models were fitted using the lme4 package [48] and analysed with type III ANOVA (with Satterthwaite’s method for the degrees of freedom estimation) from the lmerTest [49] package. Post hoc analyses were performed using the emmeans package [50], and *p*-value corrected for multiple comparisons with Tukey method. Assumptions for using a general linear model were checked with Shapiro–Wilk normality test from the rstatix package and Levene test for homoscedasticity from the car package [51], normality of the residuals’ distribution were analysed with QQ-plots (ggpubr; [52]). Where necessary, Box–Cox (BC) transformation was applied (package: MASS, [53]), which was defined as: BC(y) = (yλ-1)/λ, where λ is a value that provides the best approximation for the normal distribution of the response variable [54]. Detailed results from all the models are presented in Appendix A.

Circular statistics of the long-term MEA recordings was performed with the CircStat toolbox [55] in the MatLab Environment (R2018a version, MathWorks). Non-uniformity of distributions was confirmed with Rayleigh test and circular means were compared using the circular analogue of ANOVA (Watson-Williams test).

## 3. Results

### 3.1. Body Weight Gain and Feeding Assessment

First, we investigated how the body weight of animals changes under an HFD. In order to do so, rats were maintained on either a CD or HFD (10 animals per group) for 7 weeks and weighed every week starting at the time of weaning (week 0). We fitted a mixed effects model with diet and week as fixed factors (including the interaction between them), as well as random intercept and slope for each animal (to account for a repeated measures design of the study).

The analysis revealed significant changes in the rat body weight during the course of the study (F8,18 = 768.27, *p* < 0.0001), and differences between the diets (F1,18 = 5.08, *p* = 0.037). Due to a significant interaction (F7,18 = 16.11, *p* < 0.0001), we performed a post hoc analysis, which indicated that animals fed an HFD become heavier than the control group after 5 weeks (t18 = 2.37, *p* = 0.029), and this difference increases in the following 2 weeks (week 6: t18 = 2.87, *p* = 0.01, week 7: t18 = 3.31, *p* = 0.0039; Figure 1A).

In addition to body mass, we weighed the chow at the beginning and at the end of a day to calculate daily food intake. Rats fed an HFD were shown to eat a significantly lower amount of chow than the control group (F1,18 = 45.13, *p* < 0.0001, Figure 1B). Even though the interaction with the week of the experiment was also observed (F7,18 = 3.00, *p* = 0.028), post hoc analysis confirmed that this difference was present each week (for the results from all multiple comparisons please see Appendix A). The amount of food eaten by animals on both diets seemed to increase up until around week 3, which we believe to be due to the continuous growth of the young animals. We hypothesized, that the lower food intake of the HFD-fed group was caused by a higher caloric density of this chow; therefore, we expressed food intake as the number of calories per body mass. With this correction, the HFD-fed group was shown to consume more calories than the control group throughout the entire experiment (F1,18 = 30.52, *p* < 0.0001). Here, we also observed that the number of calories ingested per kg of body weight decreased throughout the course of the study (F7,18 = 183.84, *p* < 0.0001), with no interaction between the factors (F7,18 = 2.33, *p* = 0.07; Figure 1C).

Since our experiments were aimed at investigating changes under an HFD before the onset of obesity, it was important for us to determine how much time is needed for this to happen. The results presented in this section show that rats fed an HFD become significantly heavier than the control group after 5 weeks; therefore, all the following experiments were performed on animals fed the respective diet for a maximum of 4 weeks. Additionally, we show that rats fed an HFD eat fewer grams of food daily, but more calories per body mass, as a result of the distinct caloric density of both chows.

### 3.2. HFD Advances Daily cFos Rhythm in the DMH

Immunofluorescence staining was performed on coronal brain slices (bregma between −3.00 and −3.48, Figure 2A, [56]) collected from animals that had undergone transcardial perfusion at one of four daily time points (ZT0, 6, 12 and 18; 6 rats per group, with the exception of the HFD group perfused at ZT6 and ZT18, which contained 7 animals), 4 weeks into the experiment. Both the brains and stomachs were collected, and the latter ones immediately weighed.

Consistently with the results reported in the previous section, stomachs collected from animals fed the HFD were significantly lighter than those of the control group (F1,42 = 7.09, *p* = 0.011). Nocturnality of the rats was reflected by changes in the stomach weight between time points. For both dietary groups, the stomachs were the lightest at the beginning of the night, and the heaviest at the end of it (F3,42 = 18.98, *p* < 0.0001, Figure 2D).

Next, we analysed the changes in the number of cFos-positive neurons in the DMH, considering both the time of the day (the ZT factor) and the diet. To account for the day/night changes in stomach weight, we fitted the model including it as a covariate. The density of cFos-positive cells was defined as the number of immunostained neurons per 1 mm^2^ of the DMH area. To achieve normality of the distributions and homoscedasticity, we applied Box–Cox (BC) transformation to the data (λ = 0.59). All cFos-immunoreactive cells were also positive for DAPI (Figure 2C).

We collected and analysed a total of 140 brain slices containing the DMH (hemispheres treated separately), from 50 animals. Differences between the diets were observed (higher cFos for HFD-fed group; F1,41 = 21.16, *p* < 0.0001), as were changes around the clock, with generally lower values during the day and higher at night (ZT factor: F3,41 = 208.55, *p* < 0.0001). The cFos-positive cells appeared most densely in the ventral and medial parts of the structure, covering the area of all three previously outlined subdivisions of the DMH (ventral, dorsal and compact; Figure 2B). Most importantly, there was a significant interaction between the two factors analysed (F3,41 = 21.79, *p* < 0.0001). Whereas around and at nighttime cFos density was similar between the dietary groups, a striking difference was observed in the middle of the day when the HFD-fed group demonstrated an increased number of cFos-positive cells (ZT6, t40 = 9.17, *p* < 0.0001; Figure 2E).

The DMH is highly responsive to metabolic signals [22,23,24,25,26,27,28,29,30,31,32,33,34], which could be the reason for the observed variation in cFos-positive neurons around the clock. What is more, we recently showed that animals fed an HFD change their pattern of feeding and start eating during their normally inactive phase (daytime; [19]), which could explain an increased cFos expression in the DMH during this time of the day. However, stomach weight was shown not to influence the density of cFos-positive cells within our dataset (F1,41 = 0.083, *p* = 0.78), contradicting the presence of a positive correlation between them. This is most clearly visible at ZT0 when the stomachs are the heaviest but there are very few cFos-positive neurons in the structure (Figure 2F). Even though there appears to be some relationship between stomach weight and cFos immunoreactivity, cFos changes are better explained by the other factors in the model, predominantly the interaction between ZT and diet. Therefore, we believe the day/night rhythm in cFos immunoreactivity in the DMH, as well as its disruption under an HFD, are not simply a consequence of the animals’ changed feeding activity.

### 3.3. HFD Increases Midday Electrical Activity of the DMH

To reveal whether and how increased cFos reflects in the electrical activity of DMH neurons, we performed extracellular MEA recordings in two time points: middle of the day (~ZT6) and middle of the night (~ZT18). We recorded a total of 577 neurons from 16 animals. Since for this experiment rats had been kept on a diet for 2–4 weeks, we added the exact time of diet (in days) to the linear model as a covariate, even though it was not statistically significant (F1,4 = 1.87, *p* = 0.24).

In line with previous reports [24,35,57], the DMH was shown to possess a day/night rhythm in neuronal activity (LD: F1,5 = 14.14, *p* = 0.011), with higher firing frequency during the behaviorally active nighttime. However, an interaction with diet was also observed (F1,5 = 6.61, *p* = 0.047, Figure 3B). Post hoc analysis revealed that this day/night rhythm only concerned the DMH extracted from control animals (t9 = 4.14, *p* = 0.0022), whereas in the HFD-fed group it was completely abolished (t6 = 0.85, *p* = 0.43). Similarly to the results obtained with immunofluorescence, it was in the middle of the day that the activity of the cells in the experimental group was abnormally increased (day: t6 = 3.37, *p* = 0.017, night: t10 = 0.35, *p* = 0.73). No clear pattern in the spatial distribution of neurons firing with similar frequency was observed (Figure 3A).

### 3.4. HFD Delays Circadian Patterning of Neuronal Activity of the DMH

#### 3.4.1. Ad Libitum Feeding Experiment

The DMH is highly responsive to metabolic information arriving from the digestive system; however, it possesses an intrinsic circadian clock as well, driving spontaneous activity changes even in the absence of external synchronizers [35]. Therefore, we next aimed at investigating the influence of the HFD on the intrinsic clock properties of this structure. To do so, we performed another set of MEA recordings, starting at projected (p)ZT2 and continuing for ~30 h, which enabled us to monitor spontaneous changes in neuronal activity.

We spike-sorted a total of 390 units, out of which 264 were labelled “rhythmic” (presenting a single peak of activity within a 24 h window). For both dietary groups, the majority of the neurons were rhythmic: 70.65% (130/184) under CD and 65.05% (134/206) under HFD (three rats per group; χ21 = 1.15, *p* = 0.28). An example recording from one electrode is presented in Figure 4A, with extracted action potentials and their frequency change over time for two cells (one rhythmic and one non-rhythmic). Consistently with previous data, the mean peak time appeared during the nighttime; however, a 2 h delay was observed for the HFD group in respect to the control (all peak times presented as mean ± SD; for CD: pZT14.43 ± 1.03, for HFD: pZT16.58 ± 1.14, F1,262 = 9.82, *p* = 0.0019, Watson-Williams test; Figure 4B). No clear spatial pattern in either the presence of rhythmic cells or peak time values was observed, although for the control group the neurons with an earlier peak time seemed to appear closer to the compact part of the DMH (Figure 4C).

The results presented in the previous section (3.3. HFD increases midday electrical activity of the DMH), showing mainly an increased daytime activity of DMH neurons for the HFD-fed rats, could be explained by their daytime feeding behaviour, which might be stimulating DMH activity via a release of satiety signals. In the case of our long-term recordings, brain slice preparation started at ZT0 when both groups of animals should be fully satiated. Yet, the neuronal activity of the HFD group was still higher at pZT6 (F1,388 = 11.338, *p* = 0.00084, Figure 4D), confirming the results from the short MEA recordings and indicating that this difference is not due to a simple satiety effect from daytime food intake preceding the cull. In order to investigate possible differences in the strength of the rhythm, we also analysed the neuronal firing rate at the time of peak activity but found it unchanged by the diet (F1,3.74 = 0.019, *p* = 0.9, Figure 4E).

The presented results indicate that the neuronal activity in the DMH is the highest at night, which is in line with its involvement in feeding behaviour. However, to exclude possible phase shifts due to the procedure, we also performed a set of long-term experiments for the CD group, which started at ZT12. Even though the mean peak time value in this case appeared slightly later than previously (around ZT17.21 ± 1.15; Figure 4F), it confirmed the first half of the night as a time of the peak firing rate in this structure.

#### 3.4.2. Night-Feeding Experiment

The DMH is known to entrain to feeding schedules, participating in the anticipation of an upcoming meal [36,37,38,39,40]. Our long-term experiments on ad libitum-fed animals showed that the increased daytime activity of DMH neurons is not caused by their daytime feeding on the particular day of the experiment. However, it might be related to a repeated disturbance in the feeding behaviour, which shifted the DMH clock over a period of time. To find out whether this is true, we performed a similar set of experiments on animals that had undergone nighttime feeding (food available between ZT12–24) for 2 weeks prior to the recordings.

Night-feeding (NF) turned out to prevent the delay in peak neuronal activity of the HFD group (HFD ad libitum vs. HFD NF: F1,308 = 10.51, *p* = 0.0013, Watson-Williams test), shifting it back to pZT14.16 ± 1.21, which was not significantly different from the CD (CD ad libitum vs. HFD NF: F1,304 = 0.16, *p* = 0.69, Watson-Williams test). As expected, this feeding restriction did not cause any effect in the CD-fed animals (peak at pZT14.15 ± 1.14 for CD NF; F1,306 = 0.2, *p* = 0.65, Watson-Williams test; Figure 4B). What is more, the difference in neuronal activity at pZT6, observed between the ad libitum-fed dietary groups, was also abolished by night feeding (F1,4.92 = 2.98, *p* = 0.15, Figure 4D). Surprisingly though, NF caused an increase in the peak firing rate of the HFD-fed group (F1,352 = 5.86, *p* = 0.016, Figure 4E). This result could be related to the fact that circadian rhythms in the DMH become much stronger under many different forms of restricted feeding [37,38,39,40], and in this case it was predominantly the HFD group that was restricted since these were the animals that would otherwise consume more food during the day.

### 3.5. Rhythm Disruption in cFos Immunoreactivity and Neuronal Activity Are Prevented by Night Feeding

Following the promising results obtained on night-fed animals, indicating the possible prevention of an HFD-mediated disturbance of the DMH clock, we decided to study NF effects in more detail. For this, we repeated previous experiments on NF animals. As for the long-term recordings, they were fed ad libitum for the first 2 weeks (when no changes in the feeding pattern are yet observed between the diets [19], which was then followed by 2 weeks of NF. For these, we used a total of 16 rats, 8 per diet.

In line with the data on ad libitum-fed animals reported here, as well as previously [19], under NF, the dietary groups did not differ in body weight for the first 4 weeks either (diet: F1,14 = 0.0001, *p* = 0.99, week: F4,14 = 629.74, *p* < 0.0001, interaction: F4,14 = 1.66, *p* = 0.22; Figure 5A). The HFD-fed animals ate less grams of chow daily (F1,14 = 56.28, *p* < 0.0001; Figure 5B), but this contributed to more calories per body mass (F1,14 = 19.14, *p* = 0.00063; Figure 5C), also matching the data obtained with ad libitum-fed groups. Interestingly, total food intake (in kcal/kg) did not differ between the protocols (ad libitum vs. NF: F1,32 = 3.46, *p* = 0.07; Figure 5D), suggesting that when the animals did not have access to food during the day, they would consume their daily requirement completely during the night. This is important information, especially regarding the HFD group, which were shown to neither overeat in response to the restriction, nor stick to the amount normally ingested during the nighttime without the additional daytime feeding.

These animals were then used for the immunostaining and short-term MEA recordings in order to systematically compare the results to the ones obtained from previous experiments on ad libitum-fed rats.

For the cFos immunoreactivity analysis, we collected a total of 68 brain slices from 16 animals (which were included in the model as a random intercept). In contrast to the analysis of the ad libitum-fed animals, here we omitted stomach weight (as a covariate), as they were not collected (no PFA perfusion performed). The number of cFos-positive cells turned out to vary between day and night (F1,6 = 10.11, *p* = 0.019); however, no difference between the diets was found (F1,6 = 0.017, *p* = 0.9), nor was there an interaction between the two (F1,6 = 0.023, *p* = 0.88, Figure 5E).

As for the MEA recordings, we spike-sorted 574 neurons from the same animals (1 brain slice per animal). In this case, all rats had been on a diet for 4 weeks; therefore, in contrast to the analysis performed for the ad libitum cohort, the exact time on a diet (in days, as a covariate) was excluded from the model. Similar to the immunofluorescence, the neuronal firing rate was also observed to change between day and night (F1,12 = 5.56, *p* = 0.037) but not differ between the diets (F1,12 = 0.1, *p* = 0.75, interaction: F1,12 = 0.47, *p* = 0.51; Figure 5F).

These results confirm that the disruption of DMH daily rhythms in neuronal activity is caused by a prolonged disruption in the rhythm of food intake and can be prevented by restricted nighttime feeding.

## 4. Discussion

The presented data support the DMH involvement in the processing of metabolic information with an emphasis on its chronoregulation. First, we confirm the DMH day/night changes in the neuronal activity, observed previously [24,57], and their circadian nature [35]. Moreover, we show that an HFD can impair the observed rhythms by increasing the daytime firing rate and cFos immunoreactivity, as well as by delaying the phase of the circadian rhythm in the electrical activity of DMH neurons. Lastly and most importantly, we provide an insight into the possible prevention of such dysregulation by time-dependent feeding, restricted to the active phase of the animal.

Since this study was designed to investigate the effects of an HFD but not obesity, we started off by determining the kinetics of body weight gain for both diets and pinpointing the time necessary for them to divert. This occurred 5 weeks after the onset of the experiment (assignment into CD or HFD groups); therefore, all successive procedures were performed on animals fed either chow for a maximum of 4 weeks. For the night-feeding protocols, the animals were fed ad libitum for the first 2 weeks, followed by 2 weeks of NF, as our previous data showed that the feeding pattern changes between the dietary groups after 3 weeks [19]. Interestingly, the same study found that even though after 4 weeks of the experiment the HFD-fed group had not become significantly heavier than the control, an interaction between diet and time was observed, suggesting faster weight gain of the HFD-fed rats [19]. This was, however, not observed for our NF animals, which, together with other reports [12,13,14], indicates restricted feeding as a potent therapeutical strategy slowing down or even preventing excessive weight gain. This highlights the importance of the temporal regulation of feeding behaviour, as these animals ingest the same amount of food when fed only during the night, as they would with ad libitum access. What makes it even more striking is that animals fed an HFD consume more calories daily than the control group, yet the timing of the meals seems to matter more than this difference in its amount.

Considering the role of DMH neurons in the regulation of metabolism and food intake [58,59], it is not surprising that their peak activity is confined to the active phase of the rats (nighttime). This was true for both the intrinsic electrical firing and expression of early-response genes (cFos), possibly indicating changes in the incoming stimuli. Interestingly, this day/night rhythm is impaired by a short-term HFD lasting no more than 4 weeks. Together with our previous data [19] showing a disrupted feeding pattern under HFD, these results indicated a general metabolic dysregulation, which required further exploration. Our NF experiments clearly showed that disrupted rhythms in the DMH functioning are not a cause but an effect of an irregular food intake and can therefore be prevented by restricted nighttime feeding. Other studies have also reported the beneficial effects of time-restricted feeding, which prevents not only excessive weight gain [12,13,14], but also fat accumulation and associated inflammation, glucose intolerance and insulin resistance, as well as improves nutrient homeostasis [14].

The DMH is well known for its ability to enhance its circadian rhythm under restricted feeding [21] and entrain to different meal schedules, which has been shown for both clock gene expression and cFos immunoreactivity [36,37,38,39,40], but so far not for its electrophysiology. With our long-term MEA recordings, we were able to examine the intrinsic circadian properties of DMH neurons and found early night to be the time of their highest firing frequency. An HFD delayed that peak in activity by ~2 h, although not when the animals had been night-fed, once again confirming this to be a result of a disrupted feeding pattern. Moreover, the strength of the rhythm, indicated by the firing rate at the peak time of each individual cell, was enhanced only in the HFD NF group, as this was the group most affected by the feeding restriction that may have required the DMH clock’s adaptation.

Importantly, DMH neurons’ activity peaked in the early night, even when the slice preparation had been shifted by 10 h, negating the possible effects of the procedure itself on the circadian rhythm within the structure. Even though in this second set (starting at pZT12) the mean peak was a little bit later than the one observed for the experiments starting at pZT2, we believe this may have been caused by the immediacy of the preparation time and peak time, which might have negatively influenced the cells firing at their highest frequency at the time of the procedure, either excitotoxically killing them or at least phase shifting their rhythm.

We recently performed an extensive electrophysiological study into different subregions of the DMH, indicating important differences between its three subdivisions: compact, ventral and dorsal ([57]; Figure 2A, Figure 3A and Figure 4C). In this case, however, such a distinction did not seem reasonable. First, cFos-positive cells were the most densely observed in the ventral part, but also medially, extending through all the abovementioned subdivisions. Because of that, a separate analysis of each one of them produced the same result. Second, spatial distribution of the MEA electrodes does not allow for a clear identification of the borders between the subdivisions of the DMH; therefore, instead, we presented the results from it in a graphical form with spatial heatmaps. In line with our patch clamp study [57], the firing rate did not seem to differ between the three parts of the structure.

Interestingly though, the distribution of the rhythmic cells also did not show any pattern, despite the compact part of the DMH being recognised as the main site of its circadian clock and most pronounced in the clock gene expression [35]. We suspect this inconsistency stems from an intense network signaling within the structure, preserved in our slice preparation. The most rhythmical, compact part of the DMH might be regulating firing frequency of the cells located outside of it; however, more research into the DMH network is needed to confirm it and determine the details. Even though cells within the compact part seemed to peak earlier in the CD-ad libitum group, this trend was not clear enough to conclude from and disappeared in other groups tested, importantly including CD NF, where it should have been even more visible.

Our results indicate that HFD dysregulates feeding behaviour, which in turn impairs the DMH clock. However, the mechanisms for either of these processes remain unknown. Diet has been shown to potently influence the composition of the gut microbiome [60,61], which affects circadian rhythms in both the liver and the hypothalamus [62]. Moreover, gut microbiota composition changes depending not only on the type of ingested food but also its timing [63], suggesting some relationship to the result observed here. Germ-free mice have increased levels of two obesity-suppressing agents: BDNF in the hypothalamus and proglucagon (precursor of glucagon-like peptides—GLP1 and 2) within the digestive system [64]. Interestingly, the DMH is among the brain structures with the highest expression for the GLP1 receptor [65], and the only hypothalamic structure expressing the GLP2 receptor [66]. Moreover, the expression of the GLP1 receptor in the DMH increases in DIO [41], whereas BDNF deletion in the DMH has been shown to result in hyperphagia [67]. Therefore, we believe studying gut microbiota could prove beneficial for uncovering the mechanism responsible for the DMH clock disruption. Additionally, investigating specific subpopulations of DMH neurons, expressing different neurotransmitters, could shed some light on the pathways involved.

Regarding possible effects of DMH rhythm impairment on the development of obesity, DMH is generally considered an orexigenic structure (for a review please see [68], so its increased activity during daytime could feed back into further food intake stimulation. On the other hand, the dorsal part of the DMH connects to the autonomic nervous system, regulating metabolism and thermogenesis [58], which can influence weight gain independently of food intake. However, more research is needed to uncover whether and how the DMH clock disruption participates in the development of obesity.

In conclusion, our work presents a clear dysregulation of the DMH circadian clock under a short-term HFD. We propose that this effect is mediated by a change in the animals’ feeding behaviour and can be avoided if a healthy feeding pattern (eating only during the active phase) is kept, even without altering the amount of food ingested. Possible obesity prevention by time-restricted feeding has already been indicated elsewhere; however, to the best of our knowledge, this is the first study to prove its beneficial effects also for the hypothalamic processing including its circadian rhythmicity. This is extremely important considering the brain’s top-down control of the metabolism and feeding behaviour, clearly disrupted by an HFD even before the onset of obesity.

## Figures and Tables

**Figure 1 nutrients-14-05034-f001:**
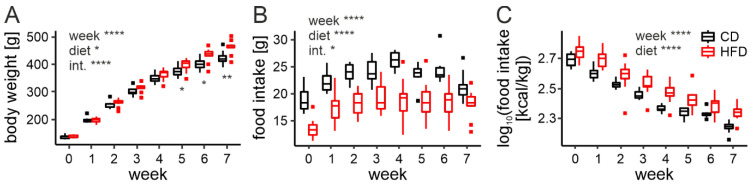
Body weight gain and feeding assessment. (**A**). Body weight gain across 7 weeks on either control (CD) or high-fat diet (HFD; week: F8,18 = 768.27, *p* < 0.0001; diet: F1,18 = 5.08, *p* = 0.037; interaction: F7,18 = 16.11, *p* < 0.0001). Differences between the diets observed from week 5 onwards (t18 = 2.37, *p* = 0.029). (**B**). Daily food intake in grams across 7 weeks on either CD or HFD (week: F7,18 = 35.92, *p* < 0.0001; diet: F1,18 = 45.13, *p* < 0.0001; interaction: F7,18 = 3.00, *p* = 0.028). (**C**). Daily food intake in kcal/kg body mass across 7 weeks on either CD or HFD (week: F7,18 = 183.84, *p* < 0.0001; diet: F1,18 = 30.52, *p* < 0.0001, interaction: F7,18 = 2.33, *p* = 0.07). * *p* < 0.05, ** *p* < 0.01, **** *p* < 0.0001. Box-and-whisker plots present median value, interquartile range (IQR; box) and the minimum-to-maximum range of values, not exceeding 1.5 * IQR (whiskers). Data points outside this range are plotted individually as outliers.

**Figure 2 nutrients-14-05034-f002:**
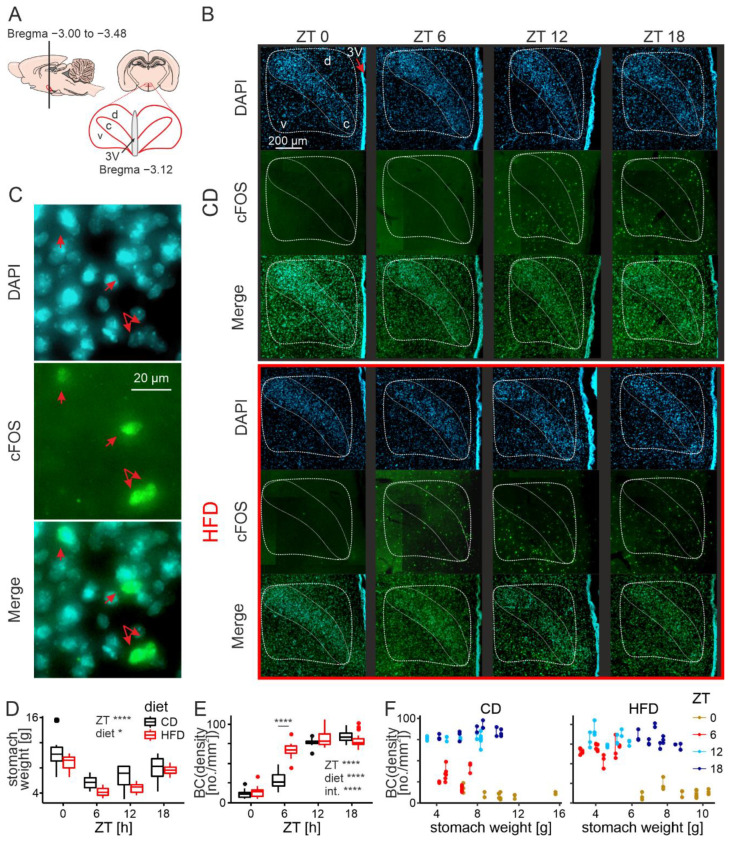
HFD advances daily cFos rhythm in the DMH. (**A**). A schematic drawing indicating the brain sections analysed in reference to bregma. (**B**). Representative microphotographs of DMH-containing brain slices obtained from animals fed either control (CD) or high-fat diet (HFD) at four daily timepoints. The compact part of the DMH, recognized with DAPI staining as the region of densely packed cells, is delineated with the white-bordered shape inside the outline of the entire structure. d—dorsal part of the DMH, c—compact part of the DMH, v—ventral part of the DMH. Please note the layer of ependymal cells indicative of the location of the third ventricle (3V) on the right side of the microphotographs. (**C**). Colocalization of DAPI and cFos within individual cells (red arrows). (**D**). Comparison of stomach weight and its changes across the 24 h between rats fed either CD or HFD (diet: F1,42 = 7.09, *p* = 0.011; ZT: F3,42 = 18.99, *p* < 0.0001; interaction: F3,42 = 0.54, *p* = 0.66). (**E**). Comparison of cFos immunoreactivity and its changes across the 24 h between rats fed either CD or HFD (diet: F1,41 = 21.16, *p* < 0.0001; ZT: F3,41 = 208.55, *p* < 0.0001; diet: F1,41 = 0.082, *p* = 0.78; interaction: F3,41 = 21.79, *p* < 0.0001). BC—Box–Cox transformed values; λ = 0.59. * *p* < 0.05, **** *p* < 0.0001. Box-and-whisker plots present the median value, the interquartile range (IQR; box) and the minimum-to-maximum range of values, not exceeding 1.5 * IQR (whiskers). Data points outside this range are plotted individually as outliers. (**F**). Graphical representation of the relationship between stomach weight and cFos-positive cell density for rats fed CD or HFD. Data points for each brain slice are presented, those acquired from the same animal are connected by a vertical line. ZT—Zeitgeber time (ZT0 at lights on).

**Figure 3 nutrients-14-05034-f003:**
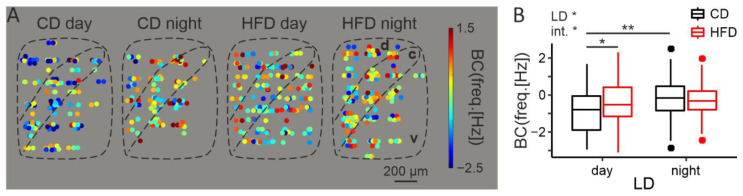
HFD increases midday electrical activity of the DMH. (**A**). Heatmaps illustrating spatial distribution of the recorded neurons and their firing frequency with color-coding. Data points from the same location (recording electrode) were scattered around it. d—dorsal part of the DMH, c—compact part of the DMH, v—ventral part of the DMH. (**B**). Comparison of the spontaneous neuronal activity between cells obtained from animals fed either control (CD) or high-fat diet (HFD), either during the day or at night (LD—light/dark; diet: F1,6 = 4, *p* = 0.096; LD: F1,5 = 14.14, *p* = 0.011, diet day: F1,4 = 1.87, *p* = 0.24; interaction: F1,5 = 6.61, *p* = 0.047). BC—Box–Cox transformed values; λ = 0.26. * *p* < 0.05, ** *p* < 0.01. Box-and-whisker plots present the median value, the interquartile range (IQR; box) and the minimum-to-maximum range of values, not exceeding 1.5 * IQR (whiskers). Data points outside this range are plotted individually as outliers.

**Figure 4 nutrients-14-05034-f004:**
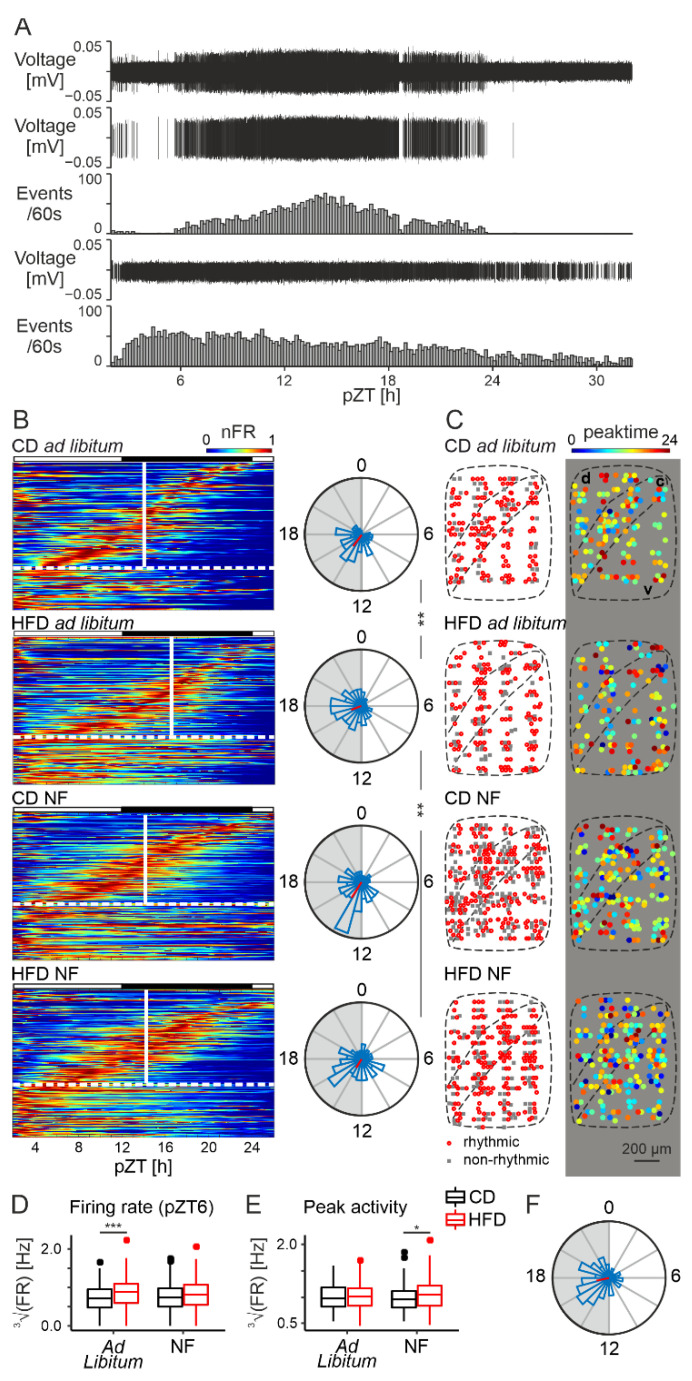
HFD delays circadian patterning of neuronal activity of the DMH. (**A**). Representative recording of a raw signal (top trace), followed by 2 spike-sorted cells (one rhythmic and one non-rhythmic), for which extracted action potentials and their frequency (extrapolated activity over 10 min) are presented. (**B**). (left) Heatmaps presenting the activity of all recorded neurons across the 24 h of recording. Cells with a single activity peak are called rhythmic and sorted by peak time. They are followed by the non-rhythmic cells, from which they are separated with a horizontal dotted line. Average peak value for each group is represented by white, vertical lines. Black and white rectangles above each heatmap reflect projected light–dark phases. (right) Rose petal diagrams (circular histograms of peak times) for each group (bin width: 1 h). The length of the vector (in red) codes the strength of the clustering. Asterisks present the results of the Watson-Williams test: ** *p* < 0.01. Shaded semi-circles represent the dark phase. CD—control diet, HFD—high-fat diet, NF—night feeding, nFR—normalized firing rate. (**C**). Heatmaps presenting spatial distribution of the rhythmic cells (left) and peak time values (right) throughout the structure. d—dorsal part of the DMH, c—compact part of the DMH, v—ventral part of the DMH. (**D**). Comparison of the firing rate at projected Zeitgeber time 6 (pZT6; Ad libitum: F1,388 = 11.338, *p* = 0.00084; NF: F1,5 = 2.98, *p* = 0.15). *** *p* < 0.001. FR—firing rate. (**E**). Comparison of the firing rate at the time of peak activity (Ad libitum: F1,4 = 0.019, *p* = 0.9; NF: F1,352 = 5.86, *p* = 0.016). * *p* < 0.05. Box-and-whisker plots present the median value, the interquartile range (IQR; box) and the minimum-to-maximum range of values, not exceeding 1.5 * IQR (whiskers). Data points outside this range are plotted individually as outliers. (**F**). Rose petal diagram (circular histogram of peaks) for an experiment starting at pZT12 (bin width: 1 h). The length of the vector (in red) codes the strength of the clustering. Shaded semi-circle represents the dark phase.

**Figure 5 nutrients-14-05034-f005:**
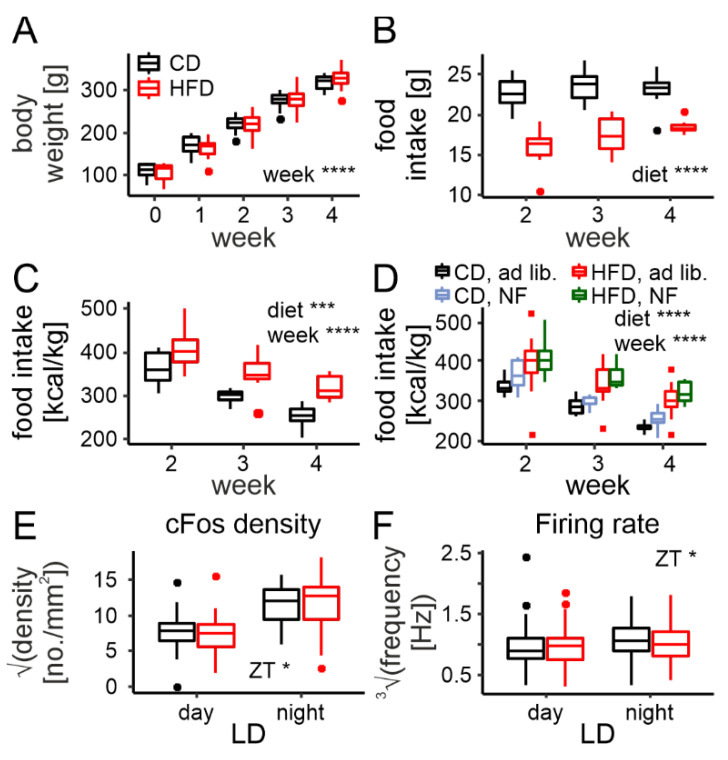
Rhythm disruption in cFos immunoreactivity and neuronal activity are prevented by night feeding (NF). (**A**). Body weight gain across 4 weeks (including 2 weeks on restricted nighttime feeding) on either control (CD) or high-fat diet (HFD; diet: F1,14 = 0.0001, *p* = 0.99, week: F4,14 = 629.74, *p* < 0.0001, interaction: F4,14 = 1.66, *p* = 0.22). (**B**). Nighttime-restricted food intake in grams for CD and HFD (diet: F1,14 = 56.28, *p* < 0.0001, week: F2,14 = 3.56, *p* = 0.056, interaction: F2,14 = 1.91, *p* = 0.18). (**C**). Nighttime-restricted food intake in kcal/kg body mass across the 2 week period (diet: F1,14 = 19.14, *p* = 0.00063, week: F2,14 = 33.71, *p* < 0.0001, interaction: F2,14 = 0.44, *p* = 0.66). (**D**). Comparison of total daily food intake between ad libitum and NF protocols for both diets (diet: F1,32 = 30.89, *p* < 0.0001, week: F2,64 = 69.14, *p* < 0.0001, protocol: F1,32 = 3.46, *p* = 0.072). (**E**). Comparison of cFos immunoreactivity and its changes between midday and midnight (LD—light/dark) between rats fed either CD or HFD during nighttime only (diet: F1,6 = 0.017, *p* = 0.9, LD: F1,6 = 10.11, *p* = 0.019, interaction: F1,6 = 0.023, *p* = 0.88. (**F**). Comparison of the neuronal firing rate and its changes between midday and midnight between rats fed either CD or HFD during nighttime only (diet: F1,12 = 0.1, *p* = 0.75, LD: F1,12 = 5.56, *p* = 0.037, interaction: F1,12 = 0.47, *p* = 0.51. * *p* < 0.05, *** *p* < 0.001, **** *p* < 0.0001. Box-and-whisker plots present the median value, the interquartile range (IQR; box) and the minimum-to-maximum range of values, not exceeding 1.5 * IQR (whiskers). Data points outside this range are plotted individually as outliers.

## Data Availability

The data presented in this study are available in Appendix A.

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
