# Peer review of "High-Fat-Diet-Evoked Disruption of the Rat Dorsomedial Hypothalamic Clock Can Be Prevented by Restricted Nighttime Feeding"

_nutrients, 2022, doi:10.3390/nu14235034_

Round 1

Reviewer 1 Report

Sanetra and colleagues studied the temporal activation of DMH neurons following HFD or HFD followed by restricted time feeding by c-Fos immunofluorescence and electrophysiology. The group discovered that 4 weeks of HFD are sufficient to disrupt the circadian rhythmicity of DMH neuron activation which can be prevented by active time-restricted feeding. The article is interested and has some strengths; however, it cannot be accepted for publication in the current form.

- Please change the wording “immunohistochemistry” with “immunofluorescence” thorough the text to be more accurate.

- Why the authors chose to study the DMH and not other hypothalamic nuclei critical for the energy balance regulation (e.g., VMH, LH, ARC)? The rationale of such choice should be motivated.

- Line 26: the statement concerning HFD duration should be more specific (2 or 4 weeks?)

- Line 109: the study was performed on 4-12 weeks rat (4 or 12 weeks? If the study initiation was 4 weeks and conclusion 12 it should be clearly stated as in this form it seems that the study was performed at both ages). Importantly, the duration of the HFD protocol should be specified in this section.

- Line 118: it is not clear the percentage of fat content which is first indicated as 42% then 70%.

- Line 224: extreme outliers’ removal should be motivated: were rats stressed, did not comply with the HFD or was there any specific reason that made them outliers? Outliers cannot be arbitrarily removed only because they are outliers.

- Line 229: type II or III ANOVA?

- Line 144: the brand and dilution of the primary antibody should be indicated.

- Line 266: please remove “in fact”.

- Figure 2: a representative Nissl staining showing where the DMH is located should be added also indicating the Bregma reference. Furthermore, the immunofluorescence images should show at least part of the third ventricle which should be indicated as 3V. The resolution of the c-Fos acquisition is not very clear: there are some lighter or darker squared areas that seem artifactual. A larger image and a larger magnification of representative areas of the ZT6 which is the timepoint showing the highest difference between groups should be added to clearly show the c-Fos positivity (and merge with DAPI) hence enabling the reader to distinguish the background from the real positivity. Such image will help understadning how the c-Fos positivity was counted. c-Fos positive cells should be only the one that show c-Fos positivity and DAPI positivity and such aspect should be specified in the method. 

- Figures 2D and E: the y axis should clearly indicate the variable. As of now, it is not clear why the authors state there is no relation between stomach weight and c-Fos staining as at ZT0 stomach weight seem to be consistently associated with lower c-Fos staining. The authors probably wanted to specify that the observed difference in c-Fos activation at ZT6 is not due to the stomach weight which is the same in both CD and HFD at this timepoint. If that is the case, such aspect should be better described as, as of now, it is not clearly in the way it is written. 

- DMH neurons activation is affected by the metabolic state and the authors chose to assess the stomach weight to correct for such variable. Is this method supported by other studies from the literature? 

- line 313: why the results are expressed as t40 and not F?

-line 463: are the authors sure about the following value referred to the diet group? F1,14= 0.00001?

- time restricted feeding protocol should be better described. Based on the way it is currently described, there are 2 weeks of ad libitum feeding (either CD or HFD?) followed by 2 weeks of night restricted feeding. It is not clear how this type of experimental design can prove that the time-restricted feeding prevents the DMH phenotype observed in the previous set of experiments: the HFD was in fact administered for a total of 4 weeks (ad libitum). To be able to compare the effect of night restricted feeding and regular HFD both protocols should be applied for the same amount of time. Alternatively, the rationale of such experimental design should be well described.

Reviewer 2 Report

The research about circadian rhythms and the impact of nutrients on it is of high interest in the field of nutrition. Hypothalamic nuclei play a key role on this control, but the mechanisms involved are not well understood. 

The work developed by the authors here presented is complementary to a previous study they have published recently (Sanetra et al, Eur J Neurosci 2022). In that paper they demonstrated that short-term high fat diet (HFD) increases the electrical activity of dorsomedial hypothalamus (DMH) neurons during the light phase. Those data agree with previous literature showing that short-HFD alters circadian rhythm and increases locomotor activity and feeding at the diurnal phase.

In the present manuscript, the authors study again the effect of short-HFD on DMH neuronal activity ex vivo. As a novelty, they also measure c-fos activity of DMH neurons at different time points of the day. In addition, they analyze the electric activity in DMH slices during 24 h confirming previous results. Finally, they demonstrate that food restriction to night phase recovers normal DMH neuron activity in HFD-fed animals similar to control diet.

Even though results are interesting and confirm that DMH activity is altered by short-HFD, this is a descriptive study that do not give information about the specific role played by DMH, whether the changes observed are cause or consequence, nor the mechanisms involved.

As a descriptive study, it should include other analysis and measurements to increase the interest and value.  Some experiments are proposed that can improve the interest and scientific soundness of the article:

- It has been described the involvement of leptin receptor positive neurons of the DMH in the control of metabolic circadian rhythm (FAber et al., eLIFE 2021). Therefore, it would be interesting to analyze whether those neurons have altered electric activity in the short-HFD model.

- Identify which type of neurons are the ones that show altered electric activity and c-fos staining upon HFD feeding.

- Analyze electrical activity of other hypothalamic nuclei such as PVH or ARC and compare the results with DMH

- Analyze food intake, food behaviour and other metabolic parameters using metabolic cages and correlate with the electric activity of DMH neurons

Reviewer 3 Report

In the present study, the authors evaluated the effect of a high-fat diet on the DMH of rats. The paper is well-written and is important in highlighting the connection between circadian rhythm and obesity. I have a few minor suggestions. 

The authors should include information on the statistics used in their figure legends for readers in addition to what they have already mentioned in the main article. 

Can the authors comment on why the food intake goes down after 3 weeks? In reference to figure 1

Can the authors speculate/discuss how weight gain is associated with increased DMH cellular activity? Can this phenotype be microbiome-dependent, as microbiome disruption due to diet affects circadian rhythms. 

Round 2

Reviewer 2 Report

The changes performed in the text and the figures have improved the quality of the manuscript.